# A Study of Catalytic Oxidation of a Library of C_2_ to C_4_ Alcohols in the Presence of Nanogold

**DOI:** 10.3390/nano9030442

**Published:** 2019-03-15

**Authors:** Maciej Kapkowski, Anna Niemczyk-Wojdyla, Piotr Bartczak, Monika Pyrkosz Bulska, Kamila Gajcy, Rafal Sitko, Maciej Zubko, Jacek Szade, Joanna Klimontko, Katarzyna Balin, Jaroslaw Polanski

**Affiliations:** 1Institute of Chemistry, University of Silesia, Szkolna 9, 40-006 Katowice, Poland; maciej.kapkowski@us.edu.pl (M.K.); anna.niemczyk@us.edu.pl (A.N.-W.); piotr.bartczak@us.edu.pl (P.B.); monika.pyrkosz-bulska@us.edu.pl (M.P.B.); kamila.gajcy@us.edu.pl (K.G.); rafal.sitko@us.edu.pl (R.S.); 2Institute of Materials Science, University of Silesia, 75 Pułku Piechoty 1A, 41-500 Chorzów, Poland; maciej.zubko@us.edu.pl; 3Department of Physics, University of Hradec Králové, Rokitanského 62, 500-03 Hradec Králové, Czech Republic; 4Institute of Physics, University of Silesia, 75 Pułku Piechoty 1A, 41-500 Chorzów, Poland; jacek.szade@us.edu.pl (J.S.); joanna.klimontko@us.edu.pl (J.K.); katarzyna.balin@us.edu.pl (K.B.)

**Keywords:** nanogold catalysis, nanomaterials, alcohol oxidation, hydrogen peroxide, silicon dioxide, green chemistry

## Abstract

The classical stoichiometric oxidation of alcohols is an important tool in contemporary organic chemistry. However, it still requires huge modifications in order to comply with the principles of green chemistry. The use of toxic chemicals, hazardous organic solvents, and the large amounts of toxic wastes that result from the reactions are a few examples of the problems that must be solved. Nanogold alone or conjugated with palladium were supported on different carriers (SiO_2_, C) and investigated in order to evaluate their catalytic potential for environmentally friendly alcohol oxidation under solvent-free and base-free conditions in the presence H_2_O_2_ as a clean oxidant. We tested different levels of Au loading (0.1–1.2% wt.) and different active catalytic site forms (monometallic Au or bimetallic Au–Pd sites). This provided new insights on how the structure of the Au-dispersions affected their catalytic performance. Importantly, the examination of the catalytic performance of the resulting catalysts was oriented toward a broad scope of alcohols, including those that are the most resistant to oxidation—the primary aliphatic alcohols. Surprisingly, the studies proved that Au/SiO_2_ at a level of Au loading as low as 0.1% wt. appeared to be efficient and prospective catalytic system for the green oxidation of alcohol. Most importantly, the results revealed that 0.1% Au/SiO_2_ might be the catalyst of choice with a wide scope of utility in the green oxidation of various structurally different alcohols as well as the non-activated aliphatic ones.

## 1. Introduction

Large amounts of alcohols can be obtained from natural and renewable sources. Therefore, they are attractive starting materials for the chemical industry. In particular, the oxidation of alcohols into their corresponding carbonyl compounds is one of the most important organic transformations because of the high value of these products in the manufacture of fine chemicals, pharmaceuticals, and special materials [1]. The classical methods of alcohol oxidation involve the use of stoichiometric amounts of toxic oxidants (such as chromates and permanganates), harmful organic solvents, and vigorous reaction conditions [2]. With the intensively growing environmental concerns and more stringent ecological standards in industry, there is an emerging quest to develop economic and efficient “green” processes for alcohol oxidation. Recently, the catalytic oxidation of alcohols that have a reusable catalytic system and environmentally benign oxidants such as O_2_ and H_2_O_2_ have received considerable attention owing to its low environmental impact, especially when compared to stoichiometric oxidation [3].

Gold-assisted catalysis is one of the fastest growing research topics in the field of catalysis, including the area of catalytic oxidation. Although gold has been regarded as being catalytically inert for centuries, studies in the 1980s revealed that nanogold particles (NPs) display an exceptional catalytic activity at low temperatures, especially in the oxidation reaction [4]. The pioneering work of Haruta et al. [4] showed that ultrafine (~5 nm) Au particles that were supported on Fe_2_O_3_, Co_3_O_4_, and NiO had a high level of activity in the low-temperature aerobic oxidation of carbon monoxide. This phenomenon has never been reached by other metals [5]. Following this outstanding finding, gold nanoparticles were used in the oxidation of alcohols into aldehydes, carboxylic acids, or esters; in the oxidation of aldehydes to esters or acids into epoxidations of olefins; and in the oxidation of amines into amides [6]. Recently, it has further been found that compared to platinum catalysts, nanogold catalysts feature higher level of activity, a higher selectivity, and a better stability for the liquid-phase oxidation of various alcohols [7]. Furthermore, the use of nanogold catalysts instead of platinum catalysts can be particularly advantageous in the case of the application of alcohol oxidation in the manufacture of active pharmaceutical ingredients (APIs). The platinum group metals are classified as “substances of significant concern”, which have residues of APIs < 10 ppm when administered orally, while gold has a limited toxicity [8]. Furthermore, gold is considerably more abundant and cheaper than the platinum group metals [9]. The unique properties of gold nanoparticles make them highly attractive in the development of novel sustainable catalytic systems for alcohol oxidation, which could be particularly suitable for industrial applications.

In the present work, a series of nanogold dispersions were prepared in order to examine their catalytic potential in the oxidation of alcohols under environmentally benign conditions. It is well known that the catalytic activity of gold catalysts is primarily determined by the particle size of Au, the properties of the support material and the method that is used to prepare the catalyst [10]. Herein, the properties of the nanogold catalysts were adapted by changing the support materials, using different levels of Au loading and incorporating a second metal in order to form a bimetallic structure.

It is well known that a number of catalysts display a high level of activity for the oxidation of specific types of alcohols [11]. For examples, Biffis et al. [12] reported that microgel-stabilized Pd nanoclusters are effective for the oxidation of secondary alcohols, while Abad et al. [13] showed that ceria-supported Au nanoparticles are most suitable for the oxidation of allylic alcohols and can prevent the isomerization and hydrogenation of the C=C double bonds.

By contrast, the aim of our studies was to find a catalytic system that can activate a wide range of substrates, including nonactivated aliphatic alcohols that can be used under mild conditions. In the model reactions, a green methodology was developed that can oxidize alcohols. In the first experiment, hydrogen peroxide was applied as the oxidant. This ‘clean’ oxidant did not produce any toxic or waste side-products and produced water as the only by-product. In the second experiment, the model reactions were carried out under solvent- and base-free conditions, in which some practical problems such as further product purification, waste base treatment, and the disposal cost of harmful wastes can be overcome.

## 2. Materials and Methods

### 2.1. Catalyst Preparation

#### 2.1.1. The Nanodispersion of Au and Pd/SiO_2_

A silica carrier was synthesized using the Stöber method with tetraethyl orthosilicate (TEOS (99.0%), Sigma Aldrich, St. Louis, MO, USA) as the silica source. The procedure was as follows: 1500 mL of anhydrous methanol (99.8%, Sigma Aldrich, St. Louis, MO, USA) and 528 mL of an ammonia solution (25 wt. %, Chempur, Piekary Śląskie, Poland) were mixed with 305 mL of deionized water, and then the mixture was stirred for 10 min. Next, 100 g of TEOS was added to the reaction mixture, which was then stirred for 5 h at room temperature. The colloidal suspension was centrifuged and then placed in an ultrasound bath and stirred for 90 min. The resulting precipitate was washed with distilled water until neutral pH was reached. In the next step, a solution of 30% HAuCl_4_ or PdCl_2_ (POCH, Gliwice, Poland), or both (Appendix A), in deionized water (10 mL) was added dropwise onto the silica carrier that had been obtained and stirred for 30 min. Then, it was dried at 60–90 °C for 12 h in the dark, ground, and sieved. Finally, without calcination a reduction of the obtained products was conducted in an oven at 500 °C for 4 h under a hydrogen atmosphere. After reduction the oven was cooled to 25 °C and purged with nitrogen for 15 min. The catalyst was stored in a gas-tight container.

#### 2.1.2. Nano-Dispersion of Au/C

The general procedure was as follows: Dispersion of 0.7% AuNPs/SiO_2_ (4.0 g) from Experiment 2.1.1 and the target carrier, i.e., C (14.0 g), were suspended in deionized water (80 mL) under mechanical stirring (BOS, Stargard, Poland) and sonication (Bandelin Sonorex, Berlin, Germany). After 10 min of vigorous stirring, sodium hydroxide (23.3 mL 40% w/w) was added to the suspension and the stirring was continued for 2 h at room temperature. Then, the suspension was allowed to stand for about 18 h until the suspended solid sedimented. In the next step, the suspension was centrifuged and the supernatant was decanted. The resulting precipitate was washed eight times with deionized water and centrifuged again to achieve neutral pH of the supernatant, which was then removed. The second generation precipitate was washed with deionized water, centrifuged, and the supernatant was removed. The catalyst that was obtained was dried in an electric dryer at 120 °C in order to get a constant weight.

### 2.2. Methods for the Characterization of the Catalyst

An energy dispersive X-ray fluorescence (EDXRF) analysis of the catalysts was performed on an Epsilon 3 spectrometer (Panalytical, Almelo, The Netherlands) with a Rh target X-ray tube that was operated at a maximum voltage of 30 keV and a maximum power of 9 W. This spectrometer is equipped with a thermoelectrically cooled silicon drift detector (SDD) that has an 8 µm Be window and a resolution of 135 eV at 5.9 keV. The quantitative analysis was performed using Omnian software and was based on the fundamental parameter method. 

The powder X-ray diffraction (XRD) experiments of the catalysts were performed on a (Panalytical, Almelo, The Netherlands) that was equipped with a pixel detector using Cu Kα radiation at 40 kV and 30 mA. The diffractograms were registered in the 10°–140° 2*θ* angle range at 0.0131° steps. A qualitative phase analysis was performed using the “X’Pert High Score Plus” computer program and the diffractograms that were obtained were compared to the standard database of the International Centre for Diffraction Data (ICDD).

The transmission electron microscopy (TEM) observations were performed using a JEOL (JEOL Ltd., Tokyo, Japan) high resolution (HRTEM) JEM 3010 microscope operating at a 300 kV accelerating voltage and equipped with a Gatan (Gatan, Inc., Pleasanton, CA, USA) 2k × 2k Orius^TM^ 833SC200D CCD camera and an EDS detector from IXRF (IXRF Systems Inc., Austin, TX, USA). The samples were suspended in isopropanol and the resulting materials were deposited on a Cu grid that had been coated with an amorphous carbon film that was standardized for TEM observations. The size distribution of the nanoparticles was calculated from the recorded TEM images. For the AuPd/SiO_2_ catalyst sample, 540 metalic nanoparticles were considered in the calculations. The concentration of nanoparticles for the 0.1% Au/SiO_2_ catalyst sample was significantly lower, and therefore only approximately 40 nanoparticles were used to determine the size distribution. Nevertheless, because the nanoparticles were more homogeneous, the lower amount of particles seems to be reasonable for obtaining good size distribution statistics.

The resulting preparations of silica or carbon-supported catalysts were examined with a Prevac/VGScienta photoelectron spectrometer (R3000 electron spectrometer, VG Scienta AB, Uppsala, Sweden and PREVAC sp. z o.o., Rogow, Poland) using X-ray photoelectron spectroscopy (XPS). Monochromatic AlKα x-ray radiation (hν = 1486.7 eV) was used to obtain the photoelectron spectra of the core levels of specific elements. The structure of the XPS multiplets that were obtained was analyzed using the Multipak program (PHI Multipak SoftwareTM Version 9.6.0.15, 2015.02.19, Ulvac-phi Inc., Chigasaki, Kanagawa, Japan) from Physical Electronics.

### 2.3. Alcohol Oxidation

Aliphatic monoalcohols (i.e., ethanol, 1-propanol, 2-propanol, and 2-butanol) as well as diols (i.e., 1,2-ethanediol, 1,2-propanediol, 1,3-propanediol, and 2,3-butanediol) were used as the model alcohols for the catalytic oxidation. The resulting nano-Au catalyst (20 mg, 0.1–1.9 μmol of Au) was suspended in a mixture of 30% H_2_O_2_ (1.0 mL, 9.8 mmol H_2_O_2_) and 0.5 mL of alcohol (1.0 mol/L, 0.5 mmol). Then, the suspension was agitated for 10 min. using a sonication bath (RK 52 H, Bandolin Electronics, 35 kHz). In the next step, the solution was stirred at 300 rpm in a sealed tube (septa system) that was placed in a thermostatic oil bath at 85 °C for 20 h. The reaction mixture was centrifuged and decanted. The supernatant was dissolved in deuterium oxide and analyzed using ^1^H and ^13^C NMR. Additionally, the 2D COSY and HMQC methods were used to identify and quantify the products. The NMR spectra were recorded on Bruker Avance 400 or Bruker Ascend 500 spectrometers (Bruker, Karlsruhe, Germany) with TMS as the internal standard (400 MHz ^1^H, 101 MHz ^13^C or 500 MHz ^1^H, 126 MHz ^13^C) at room temperature. The signal from the water was suppressed using 90 water-selective pulses. Exemplary NMR spectra with the products of the alcohol oxidation are presented in Appendix A. Equations (1)–(5) were used to calculate the conversion, product selectivity, yield, turnover number (TON), and turnover frequency (TOF), respectively.
(1)$$Conversion\;(\%)\;=\;\frac{\left(initial\;moles\;of\;alcohol\;-\;final\;moles\;of\;alcohol\right)}{initial\;moles\;of\;alcohol}\;\times \;100$$
(2)$$Selectivity\;of\;products\;(\%)\;=\;\frac{percentage\;amount\;of\;product\;formed}{the\;total\;percentage\;of\;all\;product\;formed}\;\times \;100$$
(3)$$Yield\;(\%)\;=\;\frac{conversion\;of\;alcohol\;\times \;selectivity\;of\;desired\;product}{100}$$
(4)$$TON=\frac{\alpha \cdot n_{sub}}{n_{met}}$$
(5)$$TOF=\frac{TON}{t}\;\left[h^{-1}\right]$$
where n_sub_ is the total number of moles of substrate, n_met_ is the number of moles atoms of nanometal/s, t is the time in hours, and α is the system conversion degree.

## 3. Results and Discussion

A set of five different nanogold dispersions were prepared and characterized in order to perform comparative studies on their catalytic performance in the liquid phase oxidation of alcohols. Taking into account the fact that different preparation methods can lead to different structural properties of the catalysts, we used the same technique to prepare all of the catalysts that were examined [14]. The method for preparing the catalyst was developed, proven experimentally, and previously reported [15]. The physiochemical properties of the nanogold catalysts were modified by the following changes: the type of material for the catalytic support and the level of Au-loading as well as coupling the Au with another metal in order to form bimetallic active centers instead of monometallic ones. It was assumed that a combination of these factors might affect the final properties of the resulting catalysts in many ways.

### 3.1. Characterization of the Catalysts

The EDXRF spectra of (0.2% Pd; 1.1% Au)/SiO_2_ and (1.1% Pd; 0.4% Au)/SiO_2_ are presented in Figure 1. The spectra show Au lines (Lα, Lβ, Lγ at 9.71, 11.44 and 13.38 keV, respectively), Pd lines (Lα and Lβ at 2.84 and 2.99 keV, respectively), as well as a Kα Si line at 1.74 keV. The results of the EDXRF and XPS analyses were compared with the actual values of the Au- and Pd-loading in Table 1.

For lower Au concentrations both EDXRF or XPS provide similar results which are comparable with the designed values of the Au load (Table 1, entry 1). In turn for the higher Au loads (Table 1, entries 2–5) XPS always indicated higher Au content than EDXRF. This result can be explained if we realized that X-rays (EDXRF) have a much larger penetration range compared to XPS which focuses only on the surface area. Accordingly, EDXRF relates to the bulk proportion while XPS—to the surface ratio. The surface and bulk metal to SiO_2_ ratios can take the same values only for the small amounts of the metal, when the surface portion of SiO_2_ enveloped by the metal can be neglected. Otherwise, both ratios will take a different value. The higher the percentage of the metal the higher also is the surface fraction occupied by the metal and the higher the difference between the bulk and the surface concentrations and therefore between the XPS and EDXRF analyses. The comparison of the results for pure Au and its mixtures with Pd (Table 1, entry 2 vs. entries 4 and 5) indicates that this effect is specific for each metal and/or bimetallic mixture. As the weight percentage corresponds to the bulk ratio, EDXRF correlates much better with the designed metal load. To additionally prove the above-mentioned hypothesis, we performed the Atomic Absorption Spectrometry (ASA) analyses of the 0.1% and 0.7% samples. The obtained results were 0.11 ± 0.0071 wt.% for 0.1% Au/SiO_2_ and 0.695 ± 0.0495 wt.% for 0.7% Au/SiO_2_, which compares very well with the EDXRF results. Interestingly, the only catalyst for which we obtained the same weight percentage of Au by the XPS and EDXRF analyses is the 0.1% Au/SiO_2_ system. It is also this system where Au should remain well distributed, located in the large distances between individual Au clusters. Accordingly, this should also be an optimal catalyst structure.

The XRD results of the nanogold catalysts are presented in Figure 2, which shows the X-ray diffraction patterns of the 0.7% Au/SiO_2_, 0.1% Au/SiO_2_, (1.1% Au; 0.4% Pd)/SiO_2_, and (0.2% Pd; 1.1% Au)/SiO_2_ catalysts in the range of the 2θ angle from 10 to 120 degrees. It clearly shows the diffraction lines that correspond to the pure face-centered cubic (Fm3m) phase of Au NPs (JCPDS 01-089-3697), while the considerably weaker lines of the cubic (Fm3m) phase of Pd NPs also overlaps the Au diffraction peaks. The broad peak at the low angle range is due to the silica. The Scherrer equation was used to calculate the average size of the crystalline particles. Their size was estimated from the highest intensity XRD peak (2θ_111_~38.2° for Au and 2θ_111_~39.1° for Pd NPs) and values from about 2 nm to about 10 nm were obtained. The lattice parameters (Å) of the investigated nanoparticles (calculated with the “Chekcell v.4” computer program) as well as the average crystallite dimensions (D) that were determined using the XRD method are listed in Table 2. In particular, we could observe that the 0.1% Au/SiO_2_ (Table 2, entry 1) allows one to fully control the narrow range of Au NPs size at 7 nm. This complies with the previously discussed results of the EDXRF vs. XPS analyses.

The photoelectron spectra were used to derive the atomic and weight concentrations of the main elements and to obtain information about their chemical state, including potential formation about the PdAu alloy. In particular, the concentration of Au nanoparticles on SiO_2_ carrier was found to be close to the values that were obtained from EDXRF (see Table 1) only for the lowest Au content. Taking into account the much lower escape depth of photoelectrons (up to 3–4 nm) than those of fluorescent photons, it concluded that there was a uniform distribution of nanoparticles on the surface of the core particles. The higher the concentrations of Au that was supported on SiO_2_ and also on C the higher the difference between the XPS and EDXRF values that were determined was. This might be related to the more complete coating of the silica/carbon carriers, which leads to a reduced XPS signal from the support. The same is true for the PdAu nanoparticles. The total weight concentration was higher when it was derived from the XPS spectra. The relative intensity of the Au and Pd photoemission lines permitted some conclusions to be drawn about the core–shell structure of the mixed nanoparticles. Both samples showed a similar Pd–Au weight ratio of about 1:2 and an atomic concentration close to 1:1. This may be connected to the formation of an ordered PdAu alloy mainly on the surface of the nanoparticles. The formation of such an alloy is well recognized [16,17].

A fitting of the Au 4f photoemission lines (Figure 3) confirmed the formation of the Au chemical state with a relatively low binding energy of about 83.4 eV. A similar energy level was reported for alloyed PdAu nanoparticles [17]. The analysis of the oxidation state of Pd is difficult because the most pronounced photoemission line—Pd 3d is overlapping with the stronger Au 4d one. Thus, we performed such analysis for the Pd 4p line which is relatively weak and their behavior in various chemical states is almost not present in the literature. However, we were able to fit the spectra and for both sample containing Pd we found at least two chemical states separated by a few eV. The low binding energy doublet can be assigned to PdAu alloy while the higher energy one to oxidized Pd, which is probably PdO but higher oxidation state cannot be excluded. The results of XRD (Table 2) confirmed that the alloying as the lattice constant that was derived from the Pd diffraction lines was higher than for the pure Pd for both samples, thereby indicating the formation of an alloy.

In the (1.1% Pd; 0.4% Au)/SiO_2_, the metallic nanoparticles that were distributed on the surface of SiO_2_ particles were arranged individually or as conglomerates (Figure 4a–c). The Au and Pd have the same structure (space group 227), a similar atomic radius, and their lattice parameters differed only slightly. Therefore, they created particles of solid solutions. The size of the particles had a lognormal distribution with the average particle dimensions of approximately 17 nm (Figure 4d,e). In the 0.1% Au/SiO_2_ catalyst, the gold nanoparticles were not heterogeneously distributed on the surface of the SiO_2_ particles. The particles were smaller and their size distribution could also be described using a lognormal distribution with the average particle dimensions of approximately 7 nm (Figure 4f).

### 3.2. Design and Structure of the Catalysts

When considering the carrier for nanogold dispersions, SiO_2_ and C were selected as the support for the samples of the model catalysts. The catalytic feasibility of SiO_2_ or C as a nanogold support have already been investigated in some oxidation reactions [18,19]. For example, we can refer to the work of Kapkowski et al. [18] on the efficiency of Au/SiO_2_ catalysts in glycerol oxidation using H_2_O_2_/H_2_O as a “clean” oxidant. Another good example might be the work of Carretin et al. [19] on the superior catalytic properties of a 1% wt. Au/graphite catalyst in glycerol oxidation under mild reaction conditions (60 °C, 3 h, water as the solvent). Advantageously, the alcohol oxidation in the case of both of the studies that are cited proceeded under environmentally friendly conditions. However, the application of the reported catalytic systems was limited to certain types of alcohols. Here, in contrast to the literature examples, the studies on the catalytic feasibility of the resulting nano-dispersions were extended to a broader range of alcohols.

As to the different levels of Au-loading in the resulting samples, 0.1%, 0.7% and 1.2% wt. were the nominal values. The actual values of Au-loading as measured using EDXRF analysis were consistent with the nominal ones (Table 1 vs. Appendix A). In an attempt to modify the catalytic properties by the formation of bimetallic active sites, a nano-Pd was selected as the second metal to enrich the active phase of the catalysts. It was assumed that synergistic interactions at the bimetallic active sites Au–Pd might lead to an increase in catalytic activity and stability in alcohol oxidation.

The TEM, SEM, XPS, and XRD analytical techniques were used to characterize the structure, dimensions and texture properties of the synthesized catalysts. The mean particle size of the nanometals in the active phase of the resulting catalysts was estimated by an XRD measurement (Table 2). The results proved that the mean particle size and distribution of the particle size of the resulting Au nanodispersions varied depending on the level of metal loading (Table 2, e.g., 0.1% Au/SiO_2_ vs. 0.7% Au/SiO_2_), the type of material of the support (Table 2, e.g., 0.1% Au/SiO_2_ vs. 0.2% Au/C), and the co-presence of a second metallic active phase (Table 1, e.g., (1.1% Pd; 0.4% Au)/SiO_2_ vs. (0.1% Pd; 1.1% Au)/SiO_2_ vs. 0.7% Au/SiO_2_). In all of the samples, the mean Au particle size did not exceed the critical value of 10 nm (Entries 1–5 in Table 2), which is regarded as being crucial in terms of the catalytic activity of Au. According to the literature, supported Au nanoparticles less than 10 nm in size, especially those ca. 5 nm, are typically required for catalysis [20]. It is worth noting that the resulting samples predominantly exhibited a narrow size distribution with the exception of samples 0.7% Au/SiO_2_ and (0.2% Pd; 1.1% Au)/SiO_2_ (Entries 2 and 6 in Table 2). The observed deviation from the narrow size distribution might be ascribed to a partial sintering of the Au nanoparticles for the samples that had a high level of Au-loading on the SiO_2_ support.

The model oxidation reactions of alcohols in the presence of the resulting catalysts (Table 3 and Table 4) were performed using aqueous hydrogen peroxide as the oxidant under solvent-free and base-free conditions in order to test the catalytic activity in a sustainable and environmentally benign system. When selecting the model alcohols, the criterion that the chemical reactivity of alcohols could be controlled by changing chemical structure was taken into account. Namely, a reactivity of aliphatic alcohols significantly increases in benzylic position [21]. Therefore, representatives of unactivated alcohols, namely 1-propanol and 1,2-propanediol, were used as the model alcohols in order to examine the catalytic capability for substrates that are more resistant to oxidation. A comparative evaluation of the catalytic performance was carried out taking into account the values of turnover number, turnover frequency, reaction conversion, selectivity, and yield of the main products that were obtained (Table 3 and Table 4). These parameters varied depending on the catalyst forms. The results clearly proved that using the catalytic properties of 0.1% Au/SiO_2_ to activate the conversion of the alcohols were considerably greater compared to the other catalysts (Entry 1 vs. entries 2–5 in Table 3 and Table 4). The catalytic system of 0.1% Au/SiO_2_ afforded the highest conversion of ca. 77% and 100% for the oxidation of 1-propanol and 1,2-propanodiol, respectively (Entry 1 in Table 3 and Table 4). Compared to the blank sample and the pure unsupported carriers (used as blind samples), the selectivity of the investigated reaction obviously prefers the formation of the products of oxygenation more than a direct carbonyl product, e.g., formic acid or propionic acid in 1-propanol oxidation instead direct propanol (Table 3 entries 1 vs. 6 and 7). In addition, among the catalysts that were tested, 0.1% Au/SiO_2_ had notably higher values of the turnover number (TON) and turnover frequency (TOF) (Entry 1 vs. entries 2–5 in Table 3 and Table 4). The large discrepancy in the TON or TOF values between 0.1% Au/SiO_2_ and the other catalysts confirmed that 0.1% Au/SiO_2_ exhibited the highest catalytic efficiency among the samples that were analyzed. The selectivity of the 0.1% Au/SiO_2_ system varied depending on the structure of the substrate. In particular, the catalytic oxidation of 1-proponol over 0.1% Au/SiO_2_ into acetic acid afforded a moderate selectivity ca. 57% (Entry 1 in Table 3), while the oxidation of 1,2-propanodiol in the presence of 0.1% Au/SiO_2_ resulted in a high selectivity of acetic acid ca. 95% (Entry 1 in Table 4).

The analysis of the data from Table 2, Table 3 and Table 4 offered insight on how the structure of the catalysts might affect the catalytic performance. For instance, the results confirmed that the catalytic performance of the Au/SiO_2_ system was strongly affected by the level of Au-loading. As was mentioned previously, the 0.1% Au/SiO_2_ catalyst had the highest degree of conversions among the catalysts that were used (Entry 1 in Table 3 and Table 4). Surprisingly, however, increasing the level of Au loading from 0.1% to 0.7% wt. for the Au/SiO_2_ system caused a dramatic decline in the degree of conversions (Entry 2 in Table 3 and Table 4). The oxidation over the 0.7% Au/SiO_2_ catalyst resulted in poor conversions of ca. 5% and 7% for the reactions with 1-propanol and 1,2-propanodiol, respectively. Furthermore, an analysis of the data from Table 2, Table 3 and Table 4 suggests that the catalytic activity of the Au dispersions may be sensitive to the particle size of the nano-Au as well as their size distribution. In this context, the particle size of nanometals for dispersions such as 0.7% Au/SiO_2_ or (0.2% Pd; 1.1% Au)/SiO_2_ (Entries 2 and 5 in Table 2) appeared to be insufficient to facilitate alcohol conversions (Entries 2 and 5 in Table 3 and Table 4). A further analysis of the data from Table 2 suggested that the particle size of the active phase might be affected by changes in the level of Au-loading. In this respect, a higher level of Au-loading could result in the partial sintering of the Au-particles, and subsequently could lead to a wide size distribution of the Au particles as was observed for the 0.7% Au/SiO_2_ or (0.2% Pd; 1.1% Au)/SiO_2_ (Entries 2 and 5 in Table 2). This phenomenon might also contribute to the worsening of the catalytic performance of the resulting samples. It is worth mentioning that the 0.1% Au/SiO_2_ catalyst appeared optimal, as expected from the EDXRF and XPS analyses. A possible reason for the deactivation of similar systems of AuPd alloys was ascribed by Hutchings et al. for the high Au-to-Pd ratio alloys which are especially sensitive to the high reaction temperature [22]. The results that were obtained also confirmed that the catalytic performance of the catalysts might be affected by the type of support, i.e., replacing the SiO_2_ support with a C support for the catalysts with a 0.2% wt. Au loading afforded higher conversion values and resulted in a moderate improvement of the catalytic efficiency (Entry 2 in Table 3 and Table 4 vs. Entry 3 in Table 3 and Table 4). In this context, better wettability of polar silica carrier by polar reagents can explain the difference between the SiO_2_ vs. C carrier. Moreover, the results presented in Table 3 and Table 4 indicate that oxidation depends upon many factors. In particular, paradoxically the highest conversion is observed either for the 0.1% Au/SiO_2_ or for the non-catalytic or SiO_2_ catalyzed reaction. However, it is only the catalytic 0.1% Au/SiO_2_ system where the conversion and selectivity are high enough, e.g., this can reach as much as ca. 95% AA for 1,2-propanediol (Table 4, entry 1). The individual values for 1-propanol (Table 3) or 1,2-propanediol (Table 4) compares as follows: (0.1% Au/SiO_2_ ca. 77%: Table 3, entry 1; 100%: Table 4, entry 1) vs. (none catalyst ca. 47%: Table 3, entry 7; 90%: Table 4, entry 7) vs. (none catalyst ca. 97%: Table 3, entry 6; 85%: Table 4, entry 6). To explain this effect, we should understand that the Au NPs catalyze not only the oxidation of alcohol but also the decomposition of H_2_O_2_. The latter effect is especially visible at higher temperatures. Therefore, an increasing temperature, from one side, enhances the reaction but, from the other side, enhances also the decomposition of the oxidant. In this context our previous experiments showed that 85 °C appeared more or less optimal for the process. In turn, in the non-catalytic or SiO_2_ catalyzed systems the decomposition of H_2_O_2_ is much slower, therefore, the conversion at high temperature can be still high; however, the selectivity of the reaction is much lower and the reaction yields a variety of products. As the importance of the decomposition of H_2_O_2_ increases with the increase of the metal load, therefore, also the conversions are lower when Au load increases.

The studies also enabled an examination into whether the presence of bimetallic sites in the active phase of (1.1% Pd; 0.4% Au)/SiO_2_ and (0.2% Pd; 1.1% Au)/SiO_2_ enhanced the catalytic performance. However, the conjugation of Au and Pd appeared to be less important than was expected. Although this did not afford a significant improvement of the alcohol conversions (Entries 4 and 5 in Table 3 and Table 4), a synergistic effect could be observed for 1-propanol at (0.2% Pd; 1.1% Au)/SiO_2_ where the selectivity of the acetic acid formation amounted to 100% compared to the other catalysts (Entry 4 vs. entries 1–3 and 5 in Table 3). On the other hand, the selectivity of oxidation of 1-propanol to acetic acid at the (1.1%Pd; 0.4% Au)/SiO_2_ catalyst was lower (ca. 20%), when the formation of other byproducts was promoted with the highest selectivity of ca. 79.9% compared to the other catalytic systems (Entry 5 vs. entries 1–4 in Table 3). Another example of the synergic effect between Au and the Pd alloy is that the selectivity of the formation of acetone (ca. 50%) and 1-hydroxyacetone (ca. 50%) was enhanced while the oxidation of 1,2-propanediol at (1.1% Pd; 0.4% Au)/SiO_2_ compared to the other catalytic systems (Entry 4 vs. entries 1–3 and 6 in Table 4). 

From the above comparative analyses, it can be concluded that 0.1% Au/SiO_2_ had the most advantageous catalytic performance and appeared to be the most potent catalyst among the resulting samples. Therefore, 0.1% Au/SiO_2_ was selected for further studies whose aim was to examine its catalytic utility in the oxidation of a broader spectrum of alcohols. In order to investigate the scope of alcohol oxidation with the 0.1% Au/SiO_2_–H_2_O_2_ system, the studies were extended to various structurally different alcohols. The reactions were carried out under the same experimental conditions as was the case of the previous model reactions. The results of this part of the studies are summarized in Table 5, which covers the main products of the oxidation of the alcohols. The formation of acetic acid was specifically monitored as this product could have been formed in all of the cases. The highest oxidation yields to acetic acid were observed for the dihydric alcohols, i.e., 1,2-propanediol of ca. 94.8% and 2,3-butanediol, ca. 57.6% (Entries 6 and 8 in Table 5). For the oxidation of the monoalcohols, the highest acetic acid yields were observed for 1-propanol and 2-propanol at 43.4% and 51.7%, respectively (Entries 2 and 3 in Table 5). The low nanogold content and good wettability of the carrier by polar reagents enabled the efficient use of hydrogen peroxide, thus promoting the formation of organic acids. It should also be remembered that the values of the conversion, selectivity, and yield of acetic acid and other products that were obtained varied from moderate to high depending on the alcohol substrate (Entries 1–6 in Table 5). The results confirmed that under mild reaction conditions, the 0.1% Au/SiO_2_–H_2_O_2_ system can effectively facilitate the catalytic oxidation of various nonactivated alcohols, including the most inactive primary aliphatic alcohols.

In Figure 5 we illustrated the conversion and selectivity of oxidation as a function of the ratio of 1,2-propanediol to H_2_O_2_ for 0.1% Au/SiO_2_. Milder oxidation conditions helped, to a limited extent, to avoid deep oxidative decomposition of the reactants to AA or FA acids. The possible reaction mechanism can involve two complementary routes (Scheme 1). First one comprises the C–C bond cleavage in 1,2-propanediol yielding formaldehyde and acetaldehyde (ACDE), which are further oxidized to the corresponding acids (AA or FA). In turn, a second route involves (oxy)dehydrogenation to hydroxyacetone (HYNE) and acetone (ACNE). In the last stage of oxidation, the latter two C_3_ products are oxidized to acetic acid (AA) and formic acid (FA). For the concentration of propylene glycol to H_2_O_2_/H_2_O of 1:1, 1:3, and 1:5 we observed a high fraction of C_3_ products (HYNE and ACNE), respectively. The increasing concentration of oxidant (10, 15, and 20 moles) enhanced the oxidative degradation of this reactants. For the concentration of 1:20 AA was the only product in the reaction mixture, because FA was oxidized to CO_2_. Nanogold has been extensively studied as a catalyst for glycerol, propane-1,2-diol, n-alkyl alcohol oxidation in the presence of Brönsted bases or base free conditions using oxygen and peroxides as oxidants [23,24,25]. The Au/SiO_2_ system appeared also an efficient catalyst in oxidation of cyclohexene or D-glucose [26]. Della Pina et al. described oxidation of 1,2-propanediol at 0.5% Au/TiO_2_ and 1.0% AuPd/TiO_2_ with O_2_ to lactate with acetate and formate as byproducts (conversion up to 95%). Also, benzyl alcohol can be oxidized by H_2_O_2_ in the presence of Au nanoparticles (1 nm) deposited at SBA-15 silica carrier with 96% conversion which yielded benzylic acid as a main product [23]. Dimitratos et al. obtained benzylic acid at the Au/SBA-15 catalyst suspended in the water/K_2_CO_3_ system with 96% conversion degree and 87% selectivity. In turn, the Au/C system used in catalytic oxidation of glycerol, propylene or ethylene glycol in water/sodium hydroxide yielded acidic products [24,25]. These were also the main products of our reactions (Table 5). Moreover, the conversion and selectivity to AA could be high (Figure 5). Although the oxidation of C_3_ alcohols to AA may seem unattractive, AA is an important reagent and intermediate and solvent from the industrial point of view.

The search for an efficient, versatile, and green system for the oxidation of alcohols remains a significant challenge [27]. In view of presented results, the 0.1% Au/SiO_2_ catalyst seems to have prospects for wide applications for alcohol oxidation, and notably, SiO_2_ appears to be a promising support material for nanogold particles. This is an intriguing finding taking into account the recent trends in the investigation for an optimal material for nano-Au support, which is one use of reducible metal oxides, usually Fe_3_O_4_, ZnO, CeO_2_, and TiO_2_ [28]. In fact, the interactions between the active phase and active support (e.g., Au–TiO_2_, Au–Fe_3_O_4_) via the formation of oxygen vacancies in reducible metal oxides are recognized as being one of the most effective ways to enhance the catalytic properties [29]. By contrast, SiO_2_ is a representative of the non-reducible metal oxides, which are regarded as being relatively inert materials for nano-Au support [30]. In contrast to Au/TiO_2_, which has been discussed in the most detail, Au/SiO_2_ has been minimally studied primarily due to the low activity of SiO_2_ and the difficulties in preparing catalysts [30,31]. The typical deposition method of Au on SiO_2_ might pose a problem due to the low point of the zero charge of SiO_2_ [31]; however; we have previously showed the performance of such catalysts [15]. The utility of SiO_2_ as a support for Au can be beneficial from the practical point of view and seems to provide some advantages over the reducible metal oxides. First of all, SiO_2_ has a greatly developed specific surface area that has a high porosity, which, in turn, can favor good dispersions of Au nanoparticles [32]. By contrast, TiO_2_ features a low surface area, especially after calcination, and requires further modifications to facilitate Au dispersion [33]. Most importantly, the results of our study have proven that the deposition of Au on SiO_2_ with a level of Au loading as low as 0.1% wt. led to an efficient catalytic system for the oxidation of a broad spectrum of alcohols under mild reaction conditions. To the best of our knowledge, this result is being reported for the first time. Of course, the pioneering studies of Cao’s group [34] provided proof that 1% wt. Au/TiO_2_–H_2_O_2_ was a highly effective system in Cao et al.’s [34] procedure; however; this means that the Au loading was ten times higher than the level of Au that was needed in the Au/SiO_2_–H_2_O_2_ system that is reported here. The additional advantage of SiO_2_ as the material for Au support is its relatively low price and higher chemical stability [35] compared to other materials such as TiO_2_. As to its catalytic stability, the commercially available Au/TiO_2_ tends to deteriorate during storage as it is both light- and moisture-sensitive [36]. The study of Sárkány on acetylene hydrogenation [37] demonstrated that the deactivation of Au/TiO_2_ proceeded faster than in the case Au/SiO_2_. In the studies of Masoud et al. on the selective hydrogenation of butadiene [38], the Au/SiO_2_ catalysts clearly outperformed the Au/TiO_2_ catalysts after a certain time-on-stream. In addition, SiO_2_ is generally recognized as being safe by the FDA [39,40], whereas TiO_2_ exhibits some level of toxicity [41]. This is an important advantage of using Au/SiO_2_ as a catalyst, especially in the case alcohol oxidation for pharmaceutical syntheses. 

As the fine chemical industry moves toward green and sustainable chemistry, the proposed 0.1% Au/SiO_2_–H_2_O_2_ system could be a prospective approach towards the development of an efficient catalytic nanogold platform for the oxidation of a broad spectrum of alcohols.

## 4. Conclusions

The presented results proved that the 0.1% Au/SiO_2_ catalyst has a good potential for the environmentally benign oxidation of alcohols with H_2_O_2_ in the absence of a base under organic solvent-free conditions. Notably, the 0.1% Au/SiO_2_ system exhibits sufficient activity for the oxidation of various structurally different alcohols, including those that are resistant to oxidation—primary aliphatic alcohols. At the same time under acidic conditions the Au/SiO_2_ catalyst which is unstable in the basic conditions, shows much higher durability. The paradoxical effect of the higher activity of the catalyst of the low Au load on the silica surface is also worth mentioning. In this context the catalyst quality can be monitored by the comparison of the EDXRF vs. XPS analyses.

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
