# Peer review of "A Study of Catalytic Oxidation of a Library of C2 to C4 Alcohols in the Presence of Nanogold"

_nanomaterials, 2019, doi:10.3390/nano9030442_

Author Response

Reviewer #1

The paper by Kapkowky, Polansky et al. deals with the synthesis of silica supported monometallic Au nanoparticles (Au/SiO2) and bimetallic Au&Pd/SiO2 at very low metal loading (0.1-1.2 wt%). These metal catalysts were tested in alcohols oxidation using H2O2 as oxidant; the results were compared with those of the carbon supported analogues to shed light on the role of the support. The synthetic procedure was partially reported previously by the same authors (see ref #15) and the EDXRF, WAXD and TEM techniques were used for the characterization of the size distribution of the metal nanoparticles, metal oxidation states, composition of the nanoparticles.

Some questions and comments arise from the manuscript.

1)      There is a discrepancy between the compositions determined by XPS and EDXRD (see e.g. Table 1) for both monometallic and bimetallic nanoparticles. I recommend the ICP analysis of the catalysts in order to definitively assess the metal loading. The AuNPs and Au&Pd- NPs catalyts have been extensively investigated in oxidation catalysis using gold loading in the range of 1-2wt%; it is thus suprising the poor control in the size distribution observed at 0.7 wt% of Au and the deactivation of this catalyst in the oxidation runs because of sintering of the nanoparticles. This seems mainly related to the methodology employed in this work. Please give a comment on this issue;

Thank you for these remarks. Accordingly we have explained the reasons for the differences between EDXRF and XPS results. Additionally we compared these results with the results of ASA analyses of two samples 0.1% AuSiO2 and 0.7% Au/SiO2. The structure of the catalysts revealed by the differences of the EDXRF and XPS analyses allowed us also to dicuss the differences in the individual conversion and selectivity of the reactions investigated. Accordingly, we also explained this effect.

(line 180 to 208 in the revised text):

The results of the EDXRF and XPS analyses were compared with the actual values of the Au- and Pd-loading in Table 1. For lower Au concentrations both EDXRF or XPS provide similar results which are comparable with the designed values of the Au load (Table 1, entry 1). In turn for the higher Au loads (Table 1, entries 2-5) XPS always indicated higher Au content than EDXRF. This result can be explained if we realized that X-rays (EDXRF) have a much larger penetration range compared to XPS which focuses only on the surface area. Accordingly, EDXRF relates to the bulk proportion while XPS - to the surface ratio. The surface and bulk metal to SiO2 ratios can take the same values only for the small amounts of the metal, when the surface portion of SiO2 enveloped by the metal can be neglected. Otherwise both ratios will take a different value. The higher the percentage of the metal the higher also is the surface fraction occupied by the metal and the higher the difference between the bulk and the surface concentrations and therefore between the XPS and EDXRF analyses. The comparison of the results for pure Au and its mixtures with Pd (Table 1, entry 2 vs. entry 4,5) indicates that this effect is specific for each metal and/or bimetallic mixture. As the weight percentage corresponds to the bulk ratio, EDXRF correlates much better with the designed metal load. To additionally prove the above-mentioned hypothesis, we performed the Atomic Absorption Spectrometry (ASA) analyses of the 0.1% and 0.7% samples. The obtained results were 0.11±0.0071 wt.% for 0.1% Au/SiO2 and 0.695±0.0495 wt.% for 0.7% Au/SiO2, which compares very well with the EDXRF results. Interestingly, the only catalyst for which we obtained the same weight percentage of Au by the XPS and EDXRF analyses is the 0.1% Au/SiO2 system. It is also this system where Au should remain well distributed, located in the large distances between individual Au clusters. Accordingly, this should also be an optimal catalyst structure.

(line 364 to 371 in the revised text):

In this respect, a higher level of Au-loading could result in the partial sintering of the Au-particles, and subsequently could lead to a wide size distribution of the Au particles as was observed for the 0.7% Au/SiO2 or (0.2% Pd; 1.1% Au)/SiO2 (Entries: 2 and 5 in Table 2). This phenomenon might also contribute to the worsening of the catalytic performance of the resulting samples. It is worth mentioning that the 0.1% Au/SiO2 catalyst appeared optimal, as expected from the EDXRF and XPS analyses. A possible reason for the deactivation of similar systems of Au-Pd alloys was ascribed by Hutchings et al. for the high Au to Pd ratio alloys which are especially sensitive to the high reaction temperature [22].

2)      The discussion about the formation of the Au/Pd alloy (on the surface) and the core/shell morphology is little bit confuse and misleading. On the same issue, if a single nanoparticle is expected, what is the meaning of the two sizes given in Table 2 for the AuNPs and PdNPs? What is the meaning of the range given for 0.7% Au/SiO2 in Table 2? The Scherrer method provides a single value; 

The data presented in Table 2 come from the XRD analysis which is based on fitting the peaks which were experimentally observed. In the case of the sample 0.7% Au/SiO2 the fitting with only one profile of the line was impossible. The presented two limiting values of the particle size come from the fitting with two lines with the same position of maximum and various line-width. One can expect such effect for particle size distribution.

Similarly, the two different values for Pd and Au come from the fitting results. The presence of two components within the main diffraction peaks together with the observed composition of samples may indicate to the non-uniform distribution of two metallic components within NPs and is not contradictory with formation of PdAu alloy.

3)      The authors seem forgetting that the gold catalysts have been extensively studied in glycerol, propane-1,2-diol, n-alkyl alcohol oxidation in the presence of Bronsted bases or base free conditions using oxygen as oxidant where better results in terms of both conversion and selectivity were observed (see e.g. Chem. Soc. Rev. 2012, 41, 350–369; Chem. Sci. 2012, 3, 20–44; Acc. Chem. Res. 2015, 48, 1403−1412). Please compare the results of this work with literature; 

4)      In my opinion the highest selectivity in acetic acid observed in all of oxidation runs of this work is disappointing. Aiming to obtain better selectivity in more value added reaction products I recommend to explore milder oxidation conditions avoiding oxidative degradation of the reactants;

Thank you for these remarks. Accordingly, we performed additional experiments and discussed issue 3 and 4 in a following paragraph.

(line 437 to 469 in the revised text):

In Figure 5 we illustrated the conversion an selectivity of oxidation as a function of the ratio of 1,2-propanediol to H2O2 for 0.1% Au/SiO2. Milder oxidation conditions helped, to a limited extent, to avoid deep oxidative decomposition of the reactants to AA or FA acids. The possible reaction mechanism can involve two complementary routes. First one comprises the C-C bond cleavage in 1,2-propanediol yielding formaldehyde and acetaldehyde (ACDE) which are further oxidized to the corresponding acids (AA or FA). In turn, a second route involves (oxy)dehydrogenation to hydroxyacetone (HYNE), and acetone (ACNE). In the last stage of oxidation the latter two C3 products are oxidized to acetic acid (AA) and formic acid (FA). For the concentration of propylene glycol to H2O2/H2O of 1:1, 1:3 and 1:5 we observed a high fraction of C3 products (HYNE and ACNE), respectively. The increasing concentration of oxidant (10, 15 and 20 moles) enhanced the oxidative degradation of this reactants. For the concentration of 1:20 AA was the only product in the reaction mixture, because FA was oxidized to CO2. Nanogold has been extensively studied as a catalyst for glycerol, propane-1,2-diol, n-alkyl alcohol oxidation in the presence of Brönsted bases or base free conditions using oxygen and peroxides as oxidants [23-25]. Au/SiO2 system appeared also an efficient catalyst in oxidation of cyclohexene or D-glucose [26]. Della Pina et al. described oxidation of 1,2-propanediol at 0.5% Au/TiO2 and 1.0% AuPd/TiO2 with O2 to lactate with acetate and formate as byproducts (conversion up to 95%). Also benzyl alcohol can be oxidized by H2O2 in the presence of Au nanoparticles (1 nm) deposited at SBA-15 silica carrier with 96% conversion which yielded benzylic acid as a main product [23]. Dimitratos et al. at obtained benzylic acid at the Au/SBA-15 catalyst suspended in the water/K2CO3 system with 96% conversion degree and 87% selectivity. In turn, the Au/C system used in catalytic oxidation of glycerol, propylene or ethylene glycol in water/sodium hydroxide yielded acidic products [24,25]. These were also the main products of our reactions (Table 5). Moreover, the conversion and selectivity to AA could be high (Figure 5). Although the oxidation of C3 alcohols to AA may seem unattractive, AA is an important reagent and intermediate and solvent from the industrial point of view.

Figure 5. The yields of individual products and total conversion for the reaction of 1 mol/L 1,2-propanediol and 30% H2O2/H2O. Reaction conditions: 20 mg 0.1% Au/SiO2, 85°C, 20 h, 200 rpm, 0.3 moL/L concentration 1,2-propanediol in reaction mixture after mixing the reagents. Particular colors at chart refer to: acetone (ACNE), 1-hydroxyacetone (HYNE), acetic acid (AA), formic acid (FA), acetaldehyde (ACDE) others products (OS). The experiments was conducted for the same volume of reagents (0.5 mL alcohol with 1.0 mL H2O2/H2O solution with increasing molar ratio of oxidant) in procedure described in paragraph 2.3. 

5)      On the basis of the above comments I suggest to change the emphatic title of the paper. My recommendation is to reconsider the paper for publication in Nanomaterials after that the above comments have been properly discussed.

Thank you very much for this remark. The new title is below:

A Study of Catalytic Oxidation of a Library of C2 to C4 Alcohols in the Presence of Nanogold

Reviewer 2 Report

The authors present the work on Universal Green Catalytic Nanogold Platform for the Oxidation of a Broad Spectrum of Alcohols. The subject is of interest and the manuscript is well written. Indeed, the supported gold catalysts are of great potential for future catalytic processes even in industrial scale. This work contains several good catalytic results and full characterization of studied materials are presented. This manuscript is publishable in Nanomaterials. However, some minor changes are needed. My remarks and questions are listed below:

It is not clear in the experimental section. There was no calcination step before reduction? 

Why authors choose 500°C for the reduction? This temperature is quite high for gold and Pd catalysts? 

After the reduction the catalysts were placed into the glow box? It is possible that Pd will oxidize in air. 

It would be interesting to discuss the degree of oxidation of surface Pd from XPS data. What is the real Pd-Au alloy structure? Is there some kind of surface enrichment after reduction at 500°C? 

The XPS spectra deconvolution could be improved. Au 4f peaks from metallic gold should be asymmetric. 

Table 3. It is very strange to see a huge difference in the activity of 0.1 and 0.7% Au on silica (entries 1 and 2). 0.1%Au is almost 6 times more active then 0.7%. What is the explanation? or hypotheses. 

Did authors tested author polyols such as glucose for exemple? 

Some typos are present. please correct (for example Au/SiO2-H2O instead of Au/SiO2-H2O2)

Author Response

Reviewer #2

The authors present the work on Universal Green Catalytic Nanogold Platform for the Oxidation of a Broad Spectrum of Alcohols. The subject is of interest and the manuscript is well written. Indeed, the supported gold catalysts are of great potential for future catalytic processes even in industrial scale. This work contains several good catalytic results and full characterization of studied materials are presented. This manuscript is publishable in Nanomaterials. However, some minor changes are needed. My remarks and questions are listed below:

1)      It is not clear in the experimental section. There was no calcination step before reduction? 

None calcination was needed. This information was added to the section Materials and Methods

2)      Why authors choose 500°C for the reduction? This temperature is quite high for gold and Pd catalysts? 

Such temperature allowed us for a formation of metal particles of low dimensions.

3)      After the reduction the catalysts were placed into the glow box? It is possible that Pd will oxidize in air. 

The XPS analyses proved that the Pd catalysts could be oxidized.

(line 101 to 106 in the revised text):

In the next step, a solution of 30% HAuCl4 and/or PdCl2 (Table S1, Supplementary materials) in deionized water (10 mL) was added dropwise onto the silica carrier that had been obtained and stirred for 30 min. Then, it was dried at 60–90°C for 12 h in the dark, ground and sieved. Finally, without calcination a reduction of the obtained products was conducted in an oven at 500°C for 4 h under a hydrogen atmosphere. After reduction the oven was cooled to 25°C and purged with nitrogen for 15 minutes. Catalyst was stored in a gas-tight container.

4)      It would be interesting to discuss the degree of oxidation of surface Pd from XPS data. What is the real Pd-Au alloy structure? Is there some kind of surface enrichment after reduction at 500°C? 

We changed the text in the manuscript accordingly and replaced the Fig. 3c with the new figure showing the analysis of the Pd 4p photoemission line for the sample with higher Pd content. The second sample had the same shape indicating similar chemical structure.

(line 244 to 255 in the revised text):

A fitting of the Au 4f photoemission lines (Figure 3) confirmed the formation of the Au chemical state with a relatively low binding energy of about 83.4 eV. A similar energy level was reported for alloyed PdAu nanoparticles [17]. The analysis of the oxidation state of Pd is difficult because the most pronounced photoemission line – Pd 3d is overlapping with the stronger Au 4d one. Thus, we performed such analysis for the Pd 4p line which is relatively weak and their behavior in various chemical states is almost not present in the literature. However, we were able to fit the spectra and for both sample containing Pd we found at least two chemical states separated by a few eV. The low binding energy doublet can be assigned to PdAu alloy while the higher energy one to oxidized Pd, which is probably PdO but higher oxidation state cannot be excluded. The results of XRD (Table 2) confirmed that the alloying as the lattice constant that was derived from the Pd diffraction lines was higher than for the pure Pd for both samples, thereby indicating the formation of an alloy.

5)      The XPS spectra deconvolution could be improved. Au 4f peaks from metallic gold should be asymmetric. 

We agree that one would expect an asymmetrical shape of fitted lines for Au containing NPs. Such shape was observed for Au/C, so for conducting supporting material. However, fitting with asymmetrical lines for Pd-Au sample was not successful. The reason for such behavior is hard to explain. There are some indications in the literature that the electronic structure of very small nanoparticles differs from the bulk Au and this influences the shape of the Au 4f lines.

6)      Table 3. It is very strange to see a huge difference in the activity of 0.1 and 0.7% Au on silica (entries 1 and 2). 0.1%Au is almost 6 times more active then 0.7%. What is the explanation? or hypotheses. 

The structure of the catalysts revealed by the differences of the EDXRF and XPS analyses allowed us also to discuss the differences in the individual conversion and selectivity of the reactions investigated. Accordingly, we explained this effect, as follows:

(line 180 to 208 in the revised text):

The results of the EDXRF and XPS analyses were compared with the actual values of the Au- and Pd-loading in Table 1. For lower Au concentrations both EDXRF or XPS provide similar results which are comparable with the designed values of the Au load (Table 1, entry 1). In turn for the higher Au loads (Table 1, entries 2-5) XPS always indicated higher Au content than EDXRF. This result can be explained if we realized that X-rays (EDXRF) have a much larger penetration range compared to XPS which focuses only on the surface area. Accordingly, EDXRF relates to the bulk proportion while XPS - to the surface ratio. The surface and bulk metal to SiO2 ratios can take the same values only for the small amounts of the metal, when the surface portion of SiO2 enveloped by the metal can be neglected. Otherwise both ratios will take a different value. The higher the percentage of the metal the higher also is the surface fraction occupied by the metal and the higher the difference between the bulk and the surface concentrations and therefore between the XPS and EDXRF analyses. The comparison of the results for pure Au and its mixtures with Pd (Table 1, entry 2 vs. entry 4,5) indicates that this effect is specific for each metal and/or bimetallic mixture. As the weight percentage corresponds to the bulk ratio, EDXRF correlates much better with the designed metal load. To additionally prove the above-mentioned hypothesis, we performed the Atomic Absorption Spectrometry (ASA) analyses of the 0.1% and 0.7% samples. The obtained results were 0.11±0.0071 wt.% for 0.1% Au/SiO2 and 0.695±0.0495 wt.% for 0.7% Au/SiO2, which compares very well with the EDXRF results. Interestingly, the only catalyst for which we obtained the same weight percentage of Au by the XPS and EDXRF analyses is the 0.1% Au/SiO2 system. It is also this system where Au should remain well distributed, located in the large distances between individual Au clusters. Accordingly, this should also be an optimal catalyst structure.

(line 364 to 393 in the revised text):

In this respect, a higher level of Au-loading could result in the partial sintering of the Au-particles, and subsequently could lead to a wide size distribution of the Au particles as was observed for the 0.7% Au/SiO2 or (0.2% Pd; 1.1% Au)/SiO2 (Entries: 2 and 5 in Table 2). This phenomenon might also contribute to the worsening of the catalytic performance of the resulting samples. It is worth mentioning that the 0.1% Au/SiO2 catalyst appeared optimal, as expected from the EDXRF and XPS analyses. A possible reason for the deactivation of similar systems of Au-Pd alloys was ascribed by Hutchings et al. for the high Au to Pd ratio alloys which are especially sensitive to the high reaction temperature [22]. The results that were obtained also confirmed that the catalytic performance of the catalysts might be affected by the type of support, i.e. replacing the SiO2 support with a C support for the catalysts with a 0.2% wt. Au loading afforded higher conversion values and resulted in a moderate improvement of the catalytic efficiency (Entry 2 in Tables 3 and 4 vs. Entry 3 in Tables 3 and 4). In this context better wettability of polar silica carrier by polar reagents can explain the difference between the SiO2 vs. C carrier. Moreover, the results presented in Table 3 and 4 indicate that oxidation depends upon many factors. In particular, paradoxically the highest conversion is observed either for the 0.1% Au/SiO2 or for the non-catalytic or SiO2 catalyzed reaction. However, it is only the catalytic 0.1% Au/SiO2 system where the conversion and selectivity is high enough, e.g., this can reach as much as ca. 95% AA for 1,2-propanediol (Table 4, entry 1). The individual values for 1-propanol (Table 3) or 1,2-propanediol (Table 4) compares as follows: (0.1% Au/SiO2 ca. 77%: Table 3, entry 1; 100%: Table 4, entry 1) vs. (none catalyst ca. 47%: Table 3, entry 7; 90%: Table 4, entry 7) vs. (none catalyst ca. 97%: Table 3, entry 6; 85%: Table 4, entry 6). To explain this effect we should understand that the Au NPs catalyze not only the oxidation of alcohol but also the decomposition of H2O2. The latter effect is especially visible at higher temperatures. Therefore, an increasing temperature, from one side, enhances the reaction but, from the other side, enhances also the decomposition of the oxidant. In this context our previous experiments showed that 85°C appeared more or less optimal for the process. In turn, in the non-catalytic or SiO2 catalyzed systems the decomposition of H2O2 is much slower, therefore, the conversion at high temperature can be still high, however, the selectivity of the reaction is much lower and the reaction yields a variety of products. As the importance of the decomposition of H2O2 increases with the increase of the metal load, therefore, also the conversions are lower when Au load increases.   

7)      Did authors tested author polyols such as glucose for exemple?

Thank you for this question in our previous research we investigated oxidation of cyclohexene and D-glucose we cite our research in manuscript.

(line 450 to 451 in the revised text):

Au/SiO2 system  appeared also an efficient catalyst in oxidation of cyclohexene or D-glucose [26].

8)      Some typos are present. please correct (for example Au/SiO2-H2O instead of Au/SiO2-H2O2)

Thank you for this remark, we checked manuscript and corrected mistakes.

Reviewer 3 Report

Maciej Kapkowski et al report the use of Nanogold alone or conjugated with palladium in SiO2 and  C. They investigated their catalytic potential for environmentally friendly alcohol  oxidation under solvent-free and base-free conditions in the presence H2O2 as a clean oxidant. 

The work as it is is not sufficient for novelty, since these catalysts and the oxidation reactions have been reported before. I would suggest a revision, where more attention should be given to explain the fact that without catalysts the system conversion degree is quite high. For future publication evaluation I suggest kinetic studies and the development of a mechanistic explanation.

Author Response

Reviewer #3

Maciej Kapkowski et al report the use of Nanogold alone or conjugated with palladium in SiO2 and  C. They investigated their catalytic potential for environmentally friendly alcohol  oxidation under solvent-free and base-free conditions in the presence H2O2 as a clean oxidant. 

The work as it is is not sufficient for novelty, since these catalysts and the oxidation reactions have been reported before. I would suggest a revision, where more attention should be given to explain the fact that without catalysts the system conversion degree is quite high. For future publication evaluation I suggest kinetic studies and the development of a mechanistic explanation.

The main novelty of our investigations is that basically a series of alcohols can be oxidized to acetic acid which from industrial point of view is important reagent and intermediate for further syntheses and solvent in many reactions. For example acetic acid is used the production of artificial silk, medicines (aspirin), non-flammable film and acetic essence, chloroacetic acid, carboxymethylcellulose acetate, cellulose acetate, in heating technology for descaling and as a component of buffer solutions. Acetic acid in large quantities is used as a solvent in the refining of terephthalic acid used for the large-volume production of polyethylene terephthalate (PET bottles). In the context of the above consideration catalyst which is highly selective in oxidation of many alcohols to one product can be used to oxidize the mixtures of many wasted alcohols.

The paradoxical effect of the higher activity of the catalyst of the low Au load on the silica surface is also worth mentioning. In this context the catalyst quality can be monitored by the comparison of the EDXRF vs. XPS analyses. Accordingly, we added to the text a following fragments:

(line 180 to 208 in the revised text):

The results of the EDXRF and XPS analyses were compared with the actual values of the Au- and Pd-loading in Table 1. For lower Au concentrations both EDXRF or XPS provide similar results which are comparable with the designed values of the Au load (Table 1, entry 1). In turn for the higher Au loads (Table 1, entries 2-5) XPS always indicated higher Au content than EDXRF. This result can be explained if we realized that X-rays (EDXRF) have a much larger penetration range compared to XPS which focuses only on the surface area. Accordingly, EDXRF relates to the bulk proportion while XPS - to the surface ratio. The surface and bulk metal to SiO2 ratios can take the same values only for the small amounts of the metal, when the surface portion of SiO2 enveloped by the metal can be neglected. Otherwise both ratios will take a different value. The higher the percentage of the metal the higher also is the surface fraction occupied by the metal and the higher the difference between the bulk and the surface concentrations and therefore between the XPS and EDXRF analyses. The comparison of the results for pure Au and its mixtures with Pd (Table 1, entry 2 vs. entry 4,5) indicates that this effect is specific for each metal and/or bimetallic mixture. As the weight percentage corresponds to the bulk ratio, EDXRF correlates much better with the designed metal load. To additionally prove the above-mentioned hypothesis, we performed the Atomic Absorption Spectrometry (ASA) analyses of the 0.1% and 0.7% samples. The obtained results were 0.11±0.0071 wt.% for 0.1% Au/SiO2 and 0.695±0.0495 wt.% for 0.7% Au/SiO2, which compares very well with the EDXRF results. Interestingly, the only catalyst for which we obtained the same weight percentage of Au by the XPS and EDXRF analyses is the 0.1% Au/SiO2 system. It is also this system where Au should remain well distributed, located in the large distances between individual Au clusters. Accordingly, this should also be an optimal catalyst structure.

According to reviewer's suggestion, we explained in the manuscript the reasons for high system conversion degree in blank tests and for catalyst carriers.

(line 371 to 393 in the revised text):

The results that were obtained also confirmed that the catalytic performance of the catalysts might be affected by the type of support, i.e. replacing the SiO2 support with a C support for the catalysts with a 0.2% wt. Au loading afforded higher conversion values and resulted in a moderate improvement of the catalytic efficiency (Entry 2 in Tables 3 and 4 vs. Entry 3 in Tables 3 and 4). In this context better wettability of polar silica carrier by polar reagents can explain the difference between the SiO2 vs. C carrier. Moreover, the results presented in Table 3 and 4 indicate that oxidation depends upon many factors. In particular, paradoxically the highest conversion is observed either for the 0.1% Au/SiO2 or for the non-catalytic or SiO2 catalyzed reaction. However, it is only the catalytic 0.1% Au/SiO2 system where the conversion and selectivity is high enough, e.g., this can reach as much as ca. 95% AA for 1,2-propanediol (Table 4, entry 1). The individual values for 1-propanol (Table 3) or 1,2-propanediol (Table 4) compares as follows: (0.1% Au/SiO2 ca. 77%: Table 3, entry 1; 100%: Table 4, entry 1) vs. (none catalyst ca. 47%: Table 3, entry 7; 90%: Table 4, entry 7) vs. (none catalyst ca. 97%: Table 3, entry 6; 85%: Table 4, entry 6). To explain this effect we should understand that the Au NPs catalyze not only the oxidation of alcohol but also the decomposition of H2O2. The latter effect is especially visible at higher temperatures. Therefore, an increasing temperature, from one side, enhances the reaction but, from the other side, enhances also the decomposition of the oxidant. In this context our previous experiments showed that 85°C appeared more or less optimal for the process. In turn, in the non-catalytic or SiO2 catalyzed systems the decomposition of H2O2 is much slower, therefore, the conversion at high temperature can be still high, however, the selectivity of the reaction is much lower and the reaction yields a variety of products. As the importance of the decomposition of H2O2 increases with the increase of the metal load, therefore, also the conversions are lower when Au load increases.  

We also performed additional experiments an presented new results (Figure 5).

(line 437 to 469 in the revised text):

In Figure 5 we illustrated the conversion an selectivity of oxidation as a function of the ratio of 1,2-propanediol to H2O2 for 0.1% Au/SiO2. Milder oxidation conditions helped, to a limited extent, to avoid deep oxidative decomposition of the reactants to AA or FA acids. The possible reaction mechanism can involve two complementary routes. First one comprises the C-C bond cleavage in 1,2-propanediol yielding formaldehyde and acetaldehyde (ACDE) which are further oxidized to the corresponding acids (AA or FA). In turn, a second route involves (oxy)dehydrogenation to hydroxyacetone (HYNE), and acetone (ACNE). In the last stage of oxidation the latter two C3 products are oxidized to acetic acid (AA) and formic acid (FA). For the concentration of propylene glycol to H2O2/H2O of 1:1, 1:3 and 1:5 we observed a high fraction of C3 products (HYNE and ACNE), respectively. The increasing concentration of oxidant (10, 15 and 20 moles) enhanced the oxidative degradation of this reactants. For the concentration of 1:20 AA was the only product in the reaction mixture, because FA was oxidized to CO2. Nanogold has been extensively studied as a catalyst for glycerol, propane-1,2-diol, n-alkyl alcohol oxidation in the presence of Brönsted bases or base free conditions using oxygen and peroxides as oxidants [23-25]. Au/SiO2 system appeared also an efficient catalyst in oxidation of cyclohexene or D-glucose [26]. Della Pina et al. described oxidation of 1,2-propanediol at 0.5% Au/TiO2 and 1.0% AuPd/TiO2 with O2 to lactate with acetate and formate as byproducts (conversion up to 95%). Also benzyl alcohol can be oxidized by H2O2 in the presence of Au nanoparticles (1 nm) deposited at SBA-15 silica carrier with 96% conversion which yielded benzylic acid as a main product [23]. Dimitratos et al. at obtained benzylic acid at the Au/SBA-15 catalyst suspended in the water/K2CO3 system with 96% conversion degree and 87% selectivity. In turn, the Au/C system used in catalytic oxidation of glycerol, propylene or ethylene glycol in water/sodium hydroxide yielded acidic products [24,25]. These were also the main products of our reactions (Table 5). Moreover, the conversion and selectivity to AA could be high (Figure 5). Although the oxidation of C3 alcohols to AA may seem unattractive, AA is an important reagent and intermediate and solvent from the industrial point of view.

Figure 5. The yields of individual products and total conversion for the reaction of 1 mol/L 1,2-propanediol and 30% H2O2/H2O. Reaction conditions: 20 mg 0.1% Au/SiO2, 85°C, 20 h, 200 rpm, 0.3 moL/L concentration 1,2-propanediol in reaction mixture after mixing the reagents. Particular colors at chart refer to: acetone (ACNE), 1-hydroxyacetone (HYNE), acetic acid (AA), formic acid (FA), acetaldehyde (ACDE) others products (OS). The experiments was conducted for the same volume of reagents (0.5 mL alcohol with 1.0 mL H2O2/H2O solution with increasing molar ratio of oxidant) in procedure described in paragraph 2.3.

We also improved conclusion part.

(line 508 to 517 in the revised text):

The presented results proved that the 0.1% Au/SiO2 catalyst has a good potential for the environmentally benign oxidation of alcohols with H2O2 in the absence of a base under organic solvent free-conditions. Notably, the 0.1% Au/SiO2 system exhibits a sufficient activity for the oxidation of various structurally different alcohols, including those that are resistant to oxidation – primary aliphatic alcohols. At the same time under acidic conditions the Au/SiO2 catalyst which is unstable in the basic conditions, shows much higher durability. The paradoxical effect of the higher activity of the catalyst of the low Au load on the silica surface is also worth mentioning. In this context the catalyst quality can be monitored by the comparison of the EDXRF vs. XPS analyses. The studies have demonstrated that 0.1% Au/SiO2 might be the catalyst of choice for a universal catalytic platform that is designed for the "green" oxidation of a wide range of alcohols.

Round  2

Reviewer 1 Report

I have just few further comments:

1)   Insert a footnote in Table 2 that explains the meaning of the values given for the nanoparticle size as a result of the best fit of bimodal/multimodal lineshapes;

2)   the figure 3 is distorted, please resize it;

3)   insert a reaction scheme picturing the course of the reaction that leads to the expected/observed products (line 439 to follow in page 12);

4)   The definition of TON and TOF (page 4) and the values given in Table 3 and Table 4 are misleading; with a conversion of the substrate in the range of 80-90%, how can the TON value be zero?

5)   Delete the sentence in lines 515-517 of page 14; the catalytic activity of the support (SiO2) is comparable to the one of the best nanogold catalyst.

Author Response

Reviewer #1

I have just few further comments:

1)   Insert a footnote in Table 2 that explains the meaning of the values given for the nanoparticle size as a result of the best fit of bimodal/multimodal lineshapes;

We inserted a footnote in Table 2 (lines 225- 227 in the manuscript). Now it is.

Table 2. Mean diameters (D) and lattice parameters (Å) of the nano-Au and Pd particles in the resulting catalysts as confirmed by XRD analysis.

Catalyst

Lattice   parameters [Å]

Au and/or   Pd diameters D [nm]

1

0.1%   Au/SiO2

a=4.079   (±0.005)

7.0

2

0.7%   Au/SiO2

a= 4.079   (±0.004)

2.5 – 8.5 a

3

0.2% Au/C

a=4.080   (±0.003)

9.5

4

(1.1% Pd;   0.4% Au)/SiO2

for Au a=   4.074(±0.006)

for Pd   a=4.011(±0.004)

for Au 9.0

for Pd 5.0

5

(0.2% Pd;   1.1% Au)/SiO2

for Au a=   4.080 (±0.007)

for Pd a=   4.003 (±0.006)

for Au 2.0   – 10.0 a

for Pd 5.0

a The experimental diffraction profile is a superposition of a strong narrow peak and a less intense wide line. Both lines have the same 2θ position and different values of FWHM. The given values of Au nanoparticle size correspond to the wide and narrow lines, respectively.

2)   the figure 3 is distorted, please resize it;

We resized figure 3, accordingly.

3)   insert a reaction scheme picturing the course of the reaction that leads to the expected/observed products (line 439 to follow in page 12);

Thank you very much for this remark. We added Scheme 1 (lines 440- 451 in the manuscript).

Scheme 1. Two route mechanism of oxidation or (oxy)dehydrogenation of 1,2-propanediol to acetic acid. Products in the green frame were mostly observed for the reaction mixture of 1,2-propanediol to H2O2 in the proportion of 1:1, 1:3 or 1:5. Products in the red frame were mostly observed for the reaction mixture of 1,2-propanediol to H2O2 in the proportion of 1:10, 1:15 or 1:20. The legend sitting under chemical formulas refers to the acronyms used in then text.

4)   The definition of TON and TOF (page 4) and the values given in Table 3 and Table 4 are misleading; with a conversion of the substrate in the range of 80-90%, how can the TON value be zero?

Thank you for this remark. TON value was calculated per number of moles atoms of metal/s this information is included in equation 4 at page 4 and under tables 3 and 4 in page 10. TOF is calculated TOF = TON/t [h-1]. Accordingly a value of 0 refers to a reaction without metal catalyst.

5)   Delete the sentence in lines 515-517 of page 14; the catalytic activity of the support (SiO2) is comparable to the one of the best nanogold catalyst.

Thank you, we agree. As recommended we deleted this fragment.

Reviewer 3 Report

The authors addressed aal of the questions and remarks and therefore can be published. My only suggestion is a revision os the text for minor corrections, such as the one in line  134 - izopropanol that should be isopropanol.

Author Response

Reviewer #3

The authors addressed aal of the questions and remarks and therefore can be published. My only suggestion is a revision os the text for minor corrections, such as the one in line  134 - izopropanol that should be isopropanol.

Thank you for this review. Accordingly, we carefully corrected the text, in particular, izopropanol to isopropanol in line 134.
